# Anthropogenic Influences on Soil Erosion since the Late Holocene and Contrasting Regional Sustainability in China

**Guocheng Zhang [1], Qu Chen [1,\*], Jueqi Guan [2], Guoyong Zhao [3] and Wei Wang [1]**

[1] College of Geography and Environmental Sciences, Zhejiang Normal University, Jinhua 321004, China; zhanggc@zjnu.edu.cn (G.Z.); wangwei19998@zjnu.edu.cn (W.W.)
[2] Department of Educational Technology, Zhejiang Normal University, Jinhua 321004, China; jqguan@zjnu.edu.cn
[3] School of Geographical Sciences, Xinyang Normal University, Xinyang 464000, China; zhaogy@xynu.edu.cn
\* Correspondence: chenqu@zjnu.cn

**Abstract:** A multi-disciplinary investigation of loess sections in the southeast and northwest of the Chinese Loess Plateau (CLP) and a sediment core drilled in Zhejiang Province of southeast China was conducted. Discrepancies among grain size distribution, rock magnetic properties, geochemical characteristics and chroma features, and up-section weakening relation between various proxies in the Sanmenxia loess section were found. The results were compared with those of the Baicaoyuan loess section in the northwest of the CLP and the sediment records across the plateau and elsewhere. It was suggested that human impacts began to increase soil erosion on the CLP since the middle Holocene. In addition to the increased soil erosion being decoupled from drying climate after 4 ka, renewed intensification of soil erosion was suggested to occur within the interval of 1.5–2.5 ka as a result of enhanced human activities. The two detected increases in human-induced soil erosion on the CLP are consistent with the human-driven land use changes or human–land interactions at national or regional scales, including the anthropogenic influences on the changes in the sediment load of the Yellow River. In contrast, no human impacts overwhelming hydroclimate control of soil erosion was revealed in the Beihuqiao cores, Zhejiang. The population growth during the past 2400 years showed a relative decreasing trend on CLP and a relative increasing trend in Zhejiang. It is indicated that anthropogenic factors have played a key role in modulating the Earth's surface environment. In particular, ecologically fragile areas, such as the CLP, would be much more susceptible to human disturbance and climate change. The current serious land degradation on the CLP mainly results from the negative feedback between human–land interactions. Regional heterogeneity should be taken into account for sustainable development.

**Keywords:** soil erosion; loess; Loess Plateau; Holocene; climate change; population

## 1. Introduction

Accelerated soil erosion has been pervasively changing landscapes across the world and is recognized to have substantial implications for land use sustainability. There are differences in the sensitivity to erosion among soil types and climates [1,2]. However, only recently have the studies on soil erosion at a historical scale been made and the human imprint on soil erosion been investigated [1]. The timings of the first increase in human-induced soil erosion were revealed at global and regional scales. A significant portion of the Earth's surface was indicated to shift to human-driven soil erosion following deforestation since 4 ka (thousand years ago) [3]. In southeastern Europe, impact of human activity on soil erosion was observed at 3.2 ka [4]. In China, general acceleration in sediment accumulation and human-induced soil erosion was suggested to occur at 5 ka [5]. Consistently, human activities through land-use modifications completely altered the natural vegetation trend at 5–6 ka, as revealed by a marine pollen record from the northern South China Sea [6]. By

contrast, increased soil erosion in southwestern China occurred at 1.8 ka, as weathering proxies of a core in the northern South China Sea indicated. The spatial heterogeneity remains to be further studied. It is demonstrated that soil erosion is strongly linked to land use and agricultural sustainability [7–25]. More research work on the spatial and temporal variations in soil erosion is needed in order to better understand the interactions and feedback effects.

The Chinese Loess Plateau (CLP) is known as the cradle of Chinese civilization. The loess used to serve as fertile soils upon which the great Chinese civilization was founded. However, it has become the most severe soil erosion region in the world. Soil erosion not only determined the evolution process of the Yellow River and the security of the lower reaches but also restricted the socio-economic sustainable development. Human-induced soil erosion on the CLP occurred around 2.5 or 3.1 ka, as reported [7–10]. The human impacts on accelerating sediment yields of the Yellow River over the past 2–2.5 ka [11–14] and the land use cycles on the CLP [22] were suggested. In this paper, Holocene loess records from the northwest and southeast of the CLP were studied. A sediment record in Zhejiang, southeast China was also selected for comparison. It aimed to gain a better insight into the human–land relationship within the climatic change context and the regional disparities in sustainability between the CLP and southeastern China.

## 2. Materials and Methods

The Baicaoyuan section is located in Huining County, Gansu Province, northwest CLP (Figure 1), with a mean annual precipitation of 320 mm. The information of the section and the revealed Holocene climatic evolution was reported [26]. The Sanmen Gorge area is the transition between the middle and lower reaches of the Yellow River (Figure 1). The mean annual precipitation in Sanmenxia city is approximately 550 mm. The Sanmenxia eolian section (34°47′ N, 111°16′ E, 524 m asl) is in the inlet of the Sanmen Gorge. It is totally more than 100 m thick [27]. The 4.5–22.6 m depth interval is a L1-S1 couplet, which is commonly observed within the Sanmen Gorge [28–30]. The topmost 4.5 m are Holocene paleosols, which are a distinguished light brown-red color and underlain by calcareous nodules. The Holocene paleosols can be further divided into three parts according to the color changes observed in the field: the lower weak paleosols (3.2–4.5 m), the middle developed paleosols (2.5–3.2 m), and the upper loess (0–2.5 m). Samples were collected at 2 cm interval for 0–3.2 m and at 5 cm for the rest.

Two cores (BHQ and BHQ2) were drilled at Beihuqiao village (30°22′443″ N, 119°56′237″ E) in the southeast of Hangjiahu Plain. Lacustrine and alluvial deposits are widespread, and the average annual precipitation is 1000–1400 mm in the plain. A total of 792 limnological sediment samples were collected from the 19.8 m long BHQ2 core with 2.5 cm intervals. Dating results of the sediments from both cores were attained, and the Early-middle Holocene climate was constructed based on the BHQ core [31].

Magnetic parameters of Baicaoyuan loess were measured in the Key Laboratory of Western China's Environmental Systems (Ministry of Education), Lanzhou University, using the same method used for those of Sanmenxia loess. Samples from Sanmenxia and BHQ2 were air-dried, ground, placed in diamagnetic plastic boxes, and sealed with adhesive tape. The magnetic susceptibility ($\chi$), the frequency-dependent susceptibility ($\chi_{fd}$), the anhysteretic remanent magnetisation (ARM), the ARM susceptibility ($\chi_{ARM}$), and the saturation isothermal remanent magnetisation (SIRM) were measured or acquired, as describe by Han et al. [32]. Grain size distribution was measured using a Mastersizer 2000 laser particle-size analyser. Pre-treatment of samples followed the method commonly adapted by others [33]. Major geochemical elements were measured using a Thermo Scientific ARL PERFORM'X X-ray fluorescence (XRF) sequential spectrometer [32]. Colour properties were measured with a Konica Minolta CR-400 Colour Reader. Most magnetic parameters were measured at the State Key Laboratory of Subtropical Mountain Ecology, Fujian Normal University. Major geochemical elements were measured in the Instrumental

Analysis Centre, Xinyang Normal University. The other measurements were performed in the Geography Process Laboratory, Zhejiang Normal University.

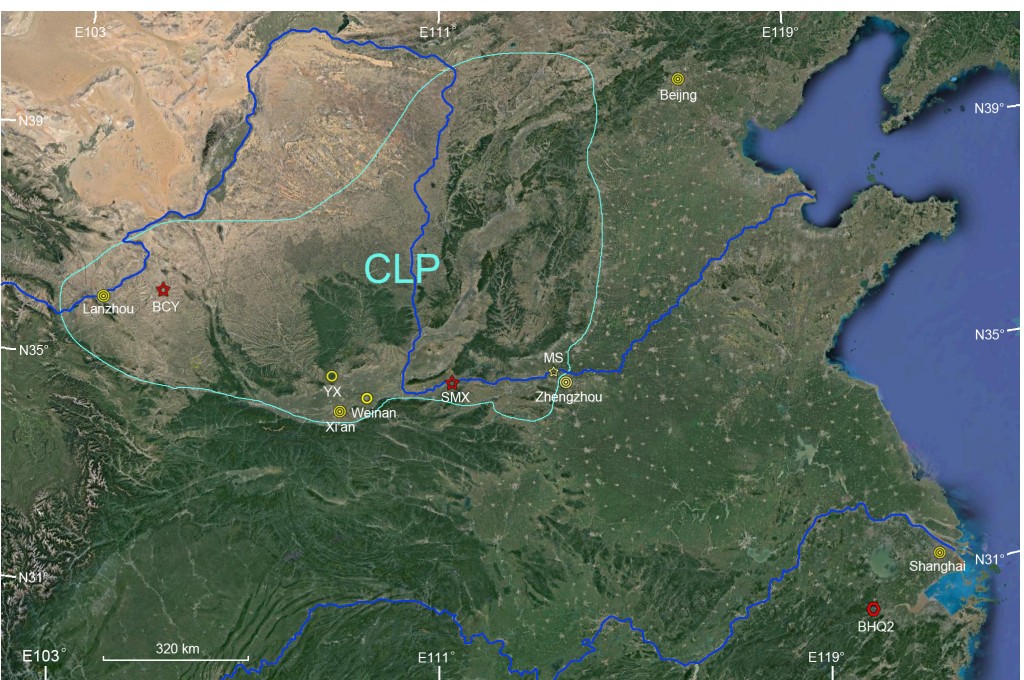

**Figure 1.** Map showing the Chinese Loess Plateau (CLP) and the location of studied sites, SMX: the Sanmenxia loess section; BCY: the Baicaoyuan loess section; MS: the Mangshan loess section; BHQ2: Beihuqiao core 2.

## 3. Results

### 3.1. Early Holocene Pedostratigraphic Correlations between Loess Sections

The results of the whole Sanmenxia section reveal a positive relation between $\chi$ and pedogenetic intensity and show minimum values of $\chi$ and $\chi_{fd}$ at around 4.5–5.5 m [27], which is consistent with the filed observation and suggests that the topmost 4.5 m are Holocene soils. The lower limit of the Holocene soils is also indicated by the correlation with the $^{14}$C dated adjacent Zhangbian loess section, which is less than 20 km away. The Zhangbian loess record since the last deglaciation was responsive to past global change [34]. Four ages, approximately 8.00 ka, 8.39 ka, 11.87 ka, and 19.49 k, were attained at 0 m, 65 m, 1.25 m, 2.65 m, and 5.55 m, respectively (Figure 2). The Younger Drays (YD) and the 8 ka cold event (also known as 8.2 ka event [26,35]) were observed in the magnetic susceptibility and grain size curves. The depth intervals at around 5.5 m and 3.2 m in the Sanmenxia section can be respectively correlated to the Younger Drays, and the 8 ka cold event. $\chi_{fd}\%$ of Sanmenxia loess is not as sensitive as grain size to the 8 ka event. However, $\chi_{fd}\%$ fluctuations at 5.5–7.3 m are similar to the GISP2 Greenland ice core record between 14.8 ka and the YD [36]. Note that the maximum of the grain size occurred later than the minimum of the magnetic parameter during the identified YD. This is consistent with the results at Zhangbian.

### 3.2. Results of Sanmenxia Loess

Redness (a*) denotes the red–green chromaticity, which is controlled mainly by the types and concentrations of iron oxides. It is sensitive to pedogenetic intensity [32]. Low values of $(CaO + Na_2O + MgO)/TiO_2$ indicate intense weathering and pedogenesis [37]. $\chi_{ARM}$ preferentially responds to stable single-domain ferrimagnetic grains. Magnetic parameters reveal the concentration, composition, and grain size of magnetic minerals [38,39]. $\chi_{ARM}/\chi$ (or ARM/SIRM) is indicative of the relative concentration of fine ferrimagnetic grains and is used as a moisture proxy. The <2 μm fraction is strongly influenced by the

pedogenetic process [40]. Coarse grain fraction content is indicative of winter monsoon intensity on the CLP. The >63 μm fraction content and median grain size (Md) are widely used as wind strength indicators [41]. The >63 μm fraction is always derived from a nearby, rather sandy source region, and is sensitive to dust storm events or cold events.

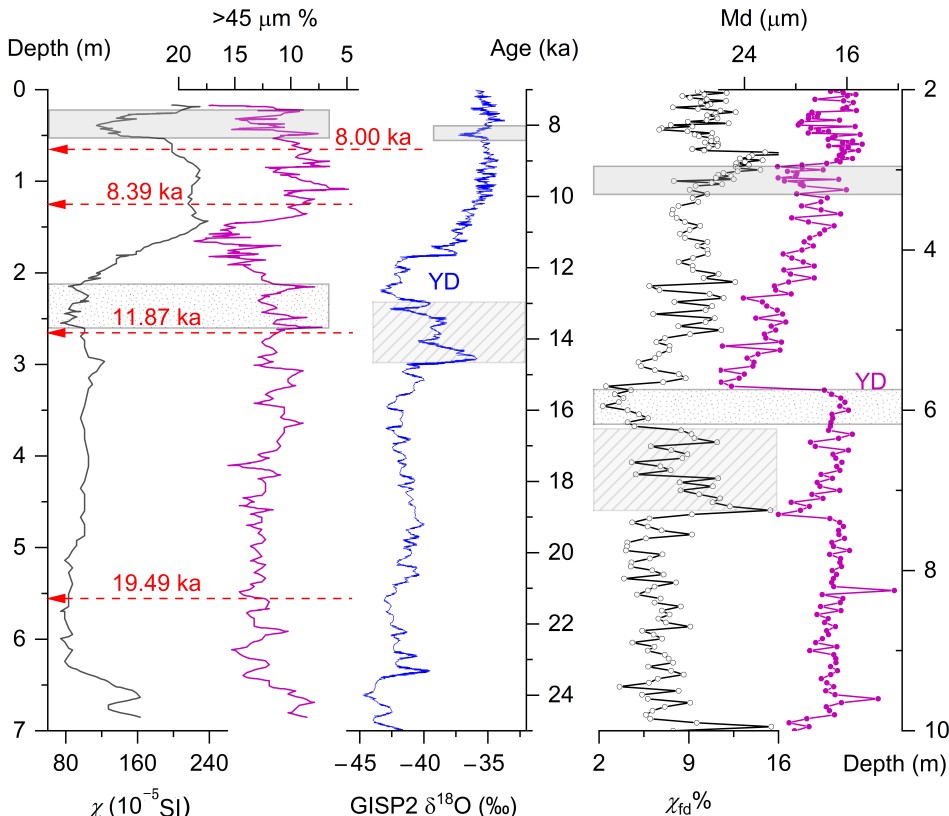

**Figure 2.** Covaries of media grain size (Md, black curve with hollow white dots) and frequency-dependent susceptibility ($\chi_{fd}$%, carmine curve with filled dots) of early Holocene loess at Sanmenxia and their correlation with records from the adjacent Zhangbian loess section (black curve for $\chi$, carmine curve for >45 μm; [34]) and the GISP2 Greenland ice core (blue curve; [36]); based on the correlation (indicated by the rectangles), the Younger Dryas (YD) is designated to the depth intervals at around 5.5 m, and 8 ka event at around 3.2 m (grey shadowed).

The redness shows an upward decreasing trend at 0–2.5 m, indicating a decrease in pedogenetic intensity. However, the geochemistry parameter (CaO + Na$_2$O + MgO)/TiO$_2$ suggests that the topmost 0.5 m has undergone more intensive chemical weathering than the 0.5–1.62 m depth interval (Figure 3). $\chi_{ARM}/\chi$ shows the more noticeable up-section changing trend at 0–3 m than the other parameters, with maximum values at 2.5–3 m and minimum values at 4.3–5.7 m and 0.1–1 m.

Consistent with field observations, all proxies suggest that the paleosols in the 1.62–3 m depth interval developed in a more humid climate compared to the neighboring layers (Figure 3). At around 2.98–3.2 m, 2.5 m, and 1.62 m, significant changes in the parameters are observed and can be correlated to the 8 ka event (Figure 1; with cold-dry conditions interpreted as occurring at 7.8 ka and 8.2 ka at Baicaoyuan [26]), 5.9 ka event [42], and 4.2 ka event [4,43]. These events were also recorded by other studied loess sections in the southeast of CPL [9,44–46]. Variations of the chromatic, geochemical and rock magnetic parameters are consistent with the identified events, as illustrated in the Section 4. However, discrepancies between various proxies can be observed.

Loess grain size distribution can help to identify the transportation dynamics and origins. The grain size parameters consistently show an up-section fining trend at 5.7–3 m and a coarsening trend at 3–0 m (Figure 3). It is worth noting that both >63 μm and <2 μm

fractions show high contents at 0–1 m. Skewness decreases with increasing grain size. The loess formed during the YD is characterized by a high concentration around a coarse modal grain size (about 40 µm) (Figure 4). In contrast, the sample from the top of the section show less concentration around the modal grain size (>30 µm), though it is also coarse and negatively skewed. Compared to the magnetic parameters, the grain size parameters have a relatively low amplitude at 0.5–3 m with less noticeable peaks at around 1.5 m.

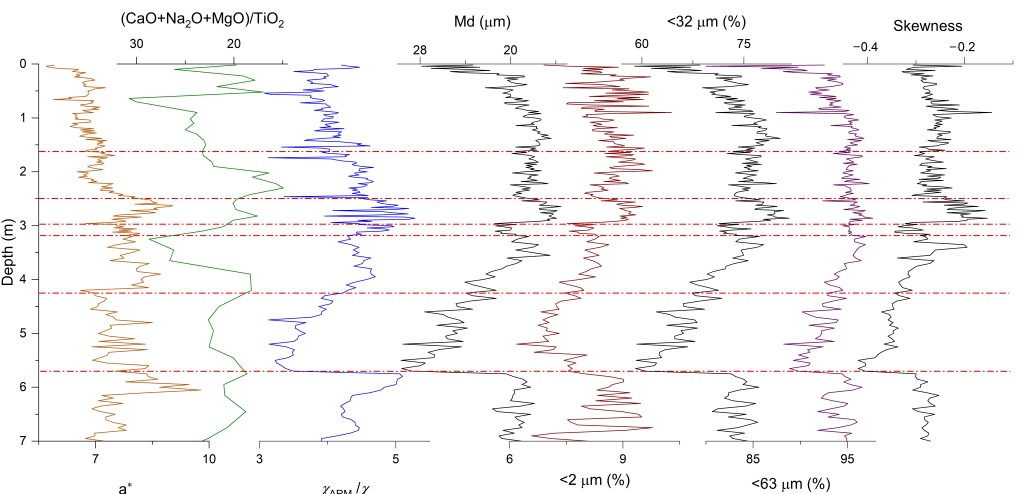

**Figure 3.** Redness (a*, yellow curve), geochemical(green curve for (CaO + Na$_2$O + MgO)/TiO$_2$), magnetic (blue curve for $\chi_{ARM}/\chi$), and granulometric parameters (left black curve for Md, red curve for <2 µm, middle black curve for <32 µm, purple curve for <63 µm, and right black curve for skewness) versus depth at Sanmenxia.

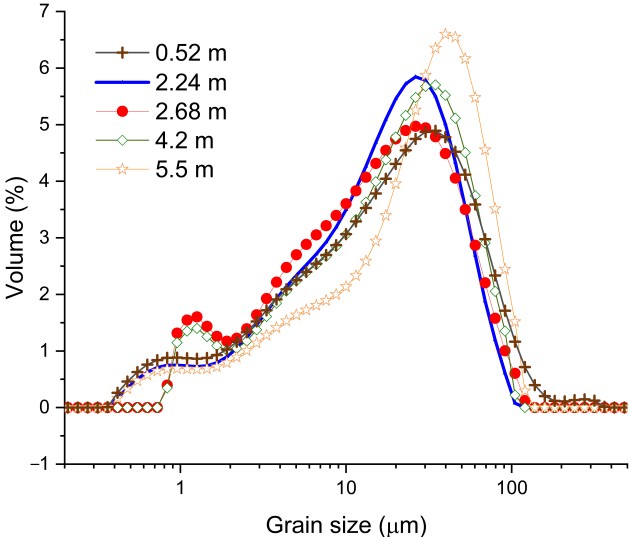

**Figure 4.** Grain size distribution of typical samples from Sanmenxia.

### 3.3. Results of BHQ2

The depth–age models of both cores drilled at Beihuqiao village are show in Figure 5, which show good agreement. It discloses that no significant change in the sediment accumulation rates occurred after 8 ka. Instead, a slight decrease in the sediment accumulation rates can be observed at 5–6 ka. The results are consistent with other sediment records in the study area [24,25].

The BHQ2 sediments are dominated by silt and clayey silt, except for a silt sand layer from 17 to 12 m (Figure 6). The core can be divided into five zones. The bottom part (below 17 m) mainly consists of dark gray clayey silt with relatively fine grain size and high $\chi$

and $\chi_{fd}$. The depth interval of 17–9 m is composed of yellow-gray and gray silt, which can be further subdivided into two layers: the lower layer dominated by gray-yellow silt (17–12 m) and the upper layer that is composed of gray silt and is relatively fine (12–9 m).

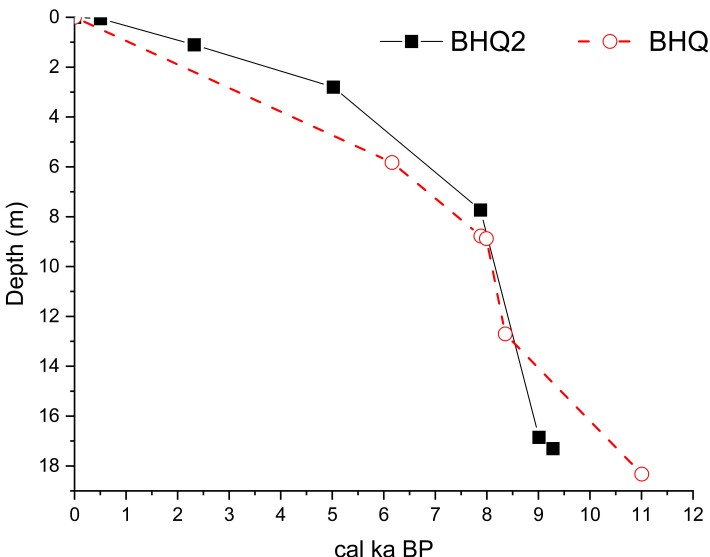

**Figure 5.** Age–depth models for BHQ and BHQ2.

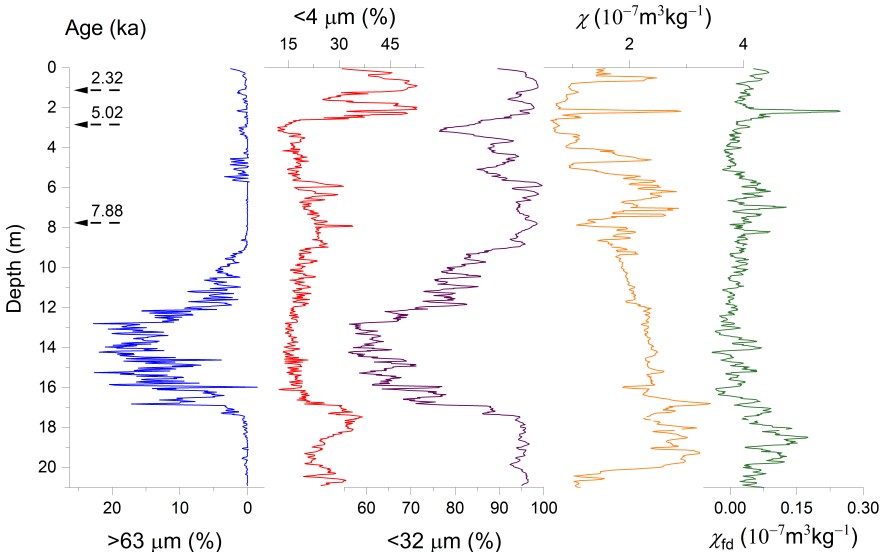

**Figure 6.** Grain size and magnetic proxies and dating results of BHQ2 (blue curve for >63 μm fraction, red curve for <4 μm fraction, purple curve for <32 μm fraction, yellow curve for $\chi$, and green curve for $\chi_{fd}$).

Moreover, the upper layer shows an upward increasing trend in the <32 μm fraction and $\chi_{fd}$ and a decreasing trend in $\chi$. The depth interval of 9–5.8 m is a dark gray and gray clayey layer with fine grain size and high $\chi$ and $\chi_{fd}$ compared to the underlying zone. The depth interval of 5.8–2.5 m shows an increase in grain size and a decrease in $\chi$ and $\chi_{fd}$. The topmost 2.5 m demonstrates a very high <4 μm fraction and a slightly high $\chi_{fd}$ compared to the underlying zone.

## 4. Discussion

### 4.1. Decoupling from Climatic Evolution since 4 ka

The relative concentration of SP (<30 nm), SSD (30~100 nm), PSD (with grain size ranging between SSD and MD), and MD (with the coarsest grain size) ferrimagnetics

(magnetite or maghemite) can disclose the origins of the magnetic imprints in loess and the provenience of loess [38,39]. $\chi_{fd}$ is sensitive only to the ferrimagnetic particles around SP and the SSD boundary, which are of postdepositional pedogenetic origin and are used as a more reliable pedogenetic or paleoclimatic proxy than $\chi$ [47]. Likewise, $\chi_{ARM}$ or ratio parameter of $\chi_{ARM}/\chi$ is used as an indicator of pedogenetic intensity [48–50].

A good relationship between $\chi$ and $\chi_{fd}$ or $\chi_{ARM}$ is found in the Sanmenxia loess, as generally observed on the Chinese Loess Plateau, suggesting that $\chi$ is dominated by ultrafine magnetic grains of pedogenetic origin resulting from an intensified summer monsoon [48–50]. The poor relationship between $\chi$ and $\chi_{fd}$ or $\chi_{ARM}$ in loess deposits can be attributed to changes in source area. In arid central Asia, $\chi_{fd}$ or $\chi_{ARM}$ decreases and shows a weak linear relation with $\chi$, and $\chi$ is positively correlated with grain size if the eolian contribution from the proximate source increases [49,50]. In southern CLP, the decrease in $\chi$ and $\chi_{fd}$ is attributed to soil erosion associated with the natural process or human disturbances [9,51]. In the Sanmenxia section under study, the stepwise decrease in $\chi$ and $\chi_{fd}$ and the weakening relation between $\chi$ versus $\chi_{fd}$ at the glacial–interglacial timescale is interpreted as being caused by soil erosion and proximate source changes for loess accumulation [27]. High SIRM in relation to $\chi$ is found in the paleosols directly overlying fluvial deposits in another loess section at Sanmenxia [52], implying that the relation between SIRM and $\chi$ or SIRM/$\chi$ is indicative of chemical weathering associated with hydrological changes.

The plots of magnetic parameters versus Md (Figure 7) and the plots of magnetic parameters (Figure 8) differentiate the depth interval at 0–2.48 m from that at 2.48–7 m in the Sanmenxia section. Furthermore, the linear relationship between SIRM and $\chi$ shows stepwise weakening at 3.25–7 m, 1–3.25 m, and 0–1 m. In contrast, the Holocene loess from Baicaoyuan, which is in the northwestern margin of the CLP and in a more arid climate, exhibits a good linear relation between the magnetic parameters (Figure 9). These observations suggest that climate alone was not adequate as the driving force of sediment dynamics and soil erosion.

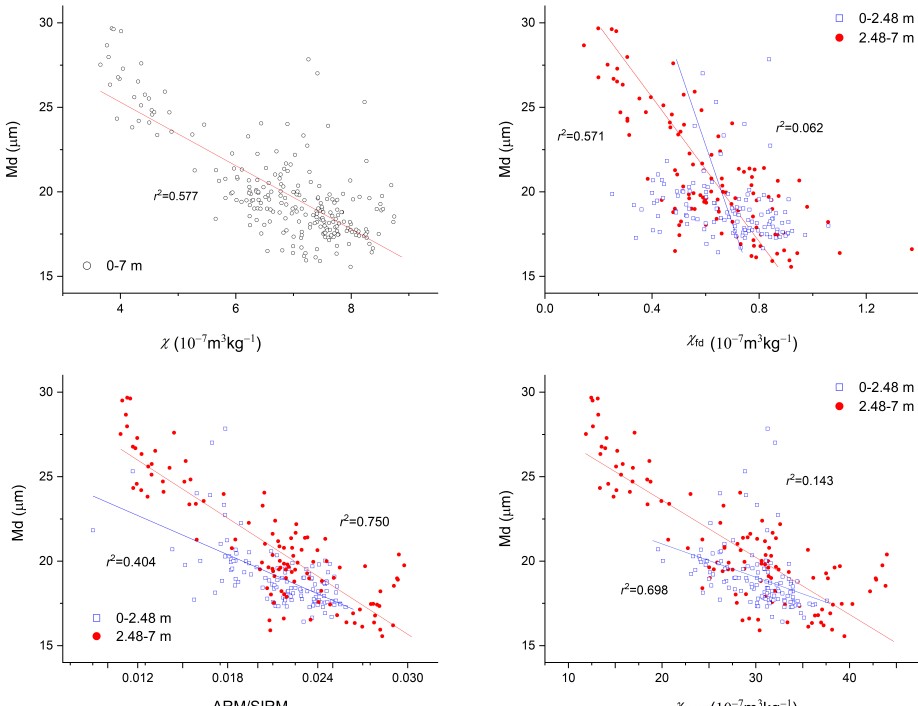

**Figure 7.** Plots of magnetic parameters versus median grain size (Md) for Sanmenxia loess.

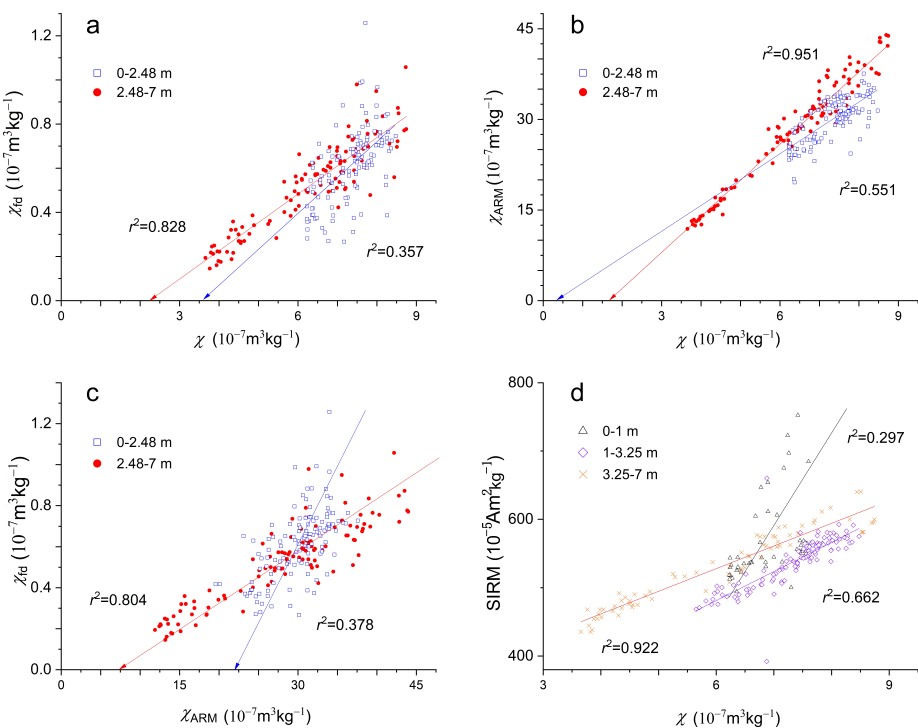

**Figure 8.** Plots of magnetic parameters for Sanmenxia loess. (**a**) plots of $\chi_{\text{fd}}$ and $\chi$, (**b**): plots of $\chi_{\text{ARM}}$ and $\chi$, (**c**) plots of $\chi_{\text{fd}}$ and $\chi_{\text{ARM}}$, (**d**) plots of SIRM and $\chi$.

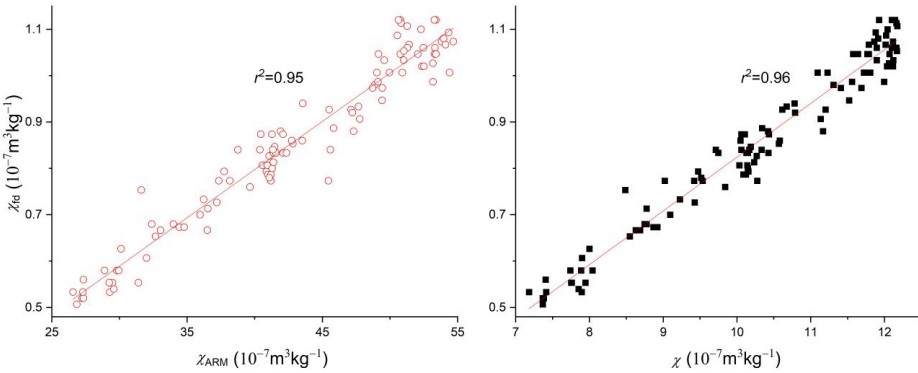

**Figure 9.** Plots of magnetic parameters for Baicaoyuan loess.

Previous studies demonstrated a similar monsoon-dominated variation pattern of the Holocene climate [53–61] and a similar magnetic enhancement model for typical loess across the CLP [48,50,54]. Based on the aforementioned fluctuations in and weakening relation between the proxies, an age frame for the Sanmenxia section is attained by interpolation with the ages of 12.9 ka, 11.7 ka, 8.2 ka, 7.8 ka, 5.9 ka, and 4.2 ka being assigned to the depth at 5.7 m, 4.25 m, 3.2 m, 2.98 m, 2.5 m, and 1.62 m, respectively (Figure 10). The Holocene paleoclimatic evolution constructed with this age frame is in concert with other results from the southeastern CLP [9,44,59,61]. The inferred paleoclimatic events are observed worldwide [45] and consistent with local records [46]. The 8 ka event is recorded by the Zhangbian section (Figure 2; [34]). The 5.9 ka event can be correlated to the ending of the Holocene optimal at 6 ka as revealed by five loess sections, including the Lingbao and Dingcun sections [58]. The 4.2 ka event can be correlated with the transition between loess subunits at Weinan and Yaoxian [9,44,62].

The reconstructed Holocene paleoclimate at Sanmenxia is consistent with the general pattern on the CLP [59] with the optimal in the middle Holocene. During 12.9–11.7 ka BP, the climate was cold and dry, dominated by the Younger Dryas. Between 11.7 and 10 ka,

the climate turned warm and wet, and 10–5.9 ka BP was the Holocene Optimum Period, during which a relatively dry-cold event at around 8 ka could be observed. At around 5.9 ka, the climate rapidly deteriorated. Afterwards, the climate slowly became dry and cold. At around 4.2 ka, the climate further deteriorated. The magnitude of paleoclimatic change is small at 4–1.5 ka but large since 1.5 ka.

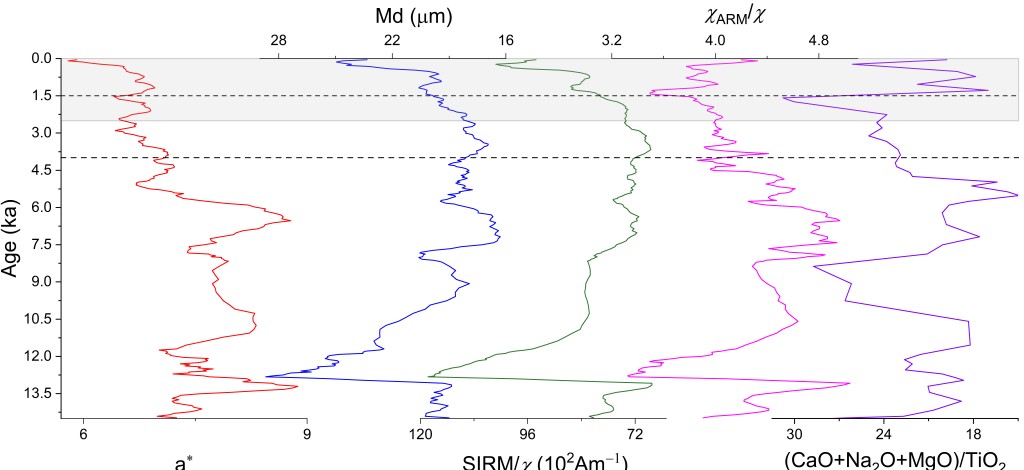

**Figure 10.** Variations of paleoclimatic proxies at Sanmenxia (red curve for a*, blue curve for Md, modena curve for $S_{300}$, carmine curve for $\chi_{ARM}/\chi$, green curve for SIRM/$\chi$, and purple curve for (CaO + Na$_2$O + MgO)/TiO$_2$). Ages are interpolated between paleoclimatic tie points at 12.9 ka (5.7 m), 11.7 ka (4.25 m), 8.2 ka (3.2 m), 7.8 ka (2.98 m), 5.9 ka (2.5 m), and 4.2 ka (1.62 m). The corresponding depths are indicated by red chain lines in Figure 3. The shadow indicates 0–2.5 ka, corresponding to the weakest relation between SIRM versus $\chi$ shown in Figure 8d. Except for the (CaO + Na$_2$O + MgO)/TiO$_2$ curve, all curves are five-point smoothed.

A recent study on Holocene loess at Yulin, a transitional zone between the CLP and Mu Us Desert, reveals asynchronous variations of East Asian summer monsoon, vegetation, and soil formation, which are ascribed to the differences in respective controlling factors [63]. It is exhibited that soil-formation-related proxies, such as magnetic susceptibility and grain size, could be inconsistent with monsoon strength. However, our results suggest that the same magnetic parameters could show different correlations at different localities (Figures 7 and 9). The result of the Sanmenxia loess shows obvious discrepancies between different paleoclimatic proxies, in particular, between magnetic proxies and grain size proxies. From 7.8–4 ka (2.98–1.5 m) to 4–1.5 ka (1.5–1 m), <2 μm%, $\chi_{ARM}/\chi$ and (CaO + Na$_2$O + MgO)/TiO$_2$ illustrated more significance changes than Md and <63 μm (Figures 3 and 10). Since 1.5 ka (1 m), the grain size coarsened, and the hard magnetic mineral content might increase (as revealed by SIRM/$\chi$ in Figure 10). Such observation is generally interpreted as resulting from the drying climate. However, (CaO + Na$_2$O + MgO)/TiO$_2$ showed intensified chemical weathering since 1.5 ka (Figures 3 and 10). Therefore SIRM/$\chi$ as well as the geochemistry parameter probably indicate accelerated soil erosion [19,52].

It is suggested that the paleoclimatic evolution in east China since 4 ka was characterized by 13 low-amplitude warm/cold fluctuations [64] or a gradual cooling trend [45,62]. A recent study in south CLP suggests that the rainfall decreased after 5 ka [53] (Figure 11). Newly synthesized results suggest that sediment accumulation rates covaried with monsoon intensity during 40–6 ka, increasing/decreasing with intensifying/weakening monsoons but increased rapidly and decoupled from weakened hydroclimates after 5 ka [5] (Figure 11). Consistently, the weakening relation between various parameters after 4.2 ka (above 2.48 m) and the discrepancy between various proxies at Sanmenxia could be interpreted as being caused by other factors that overwhelmed climatic control.

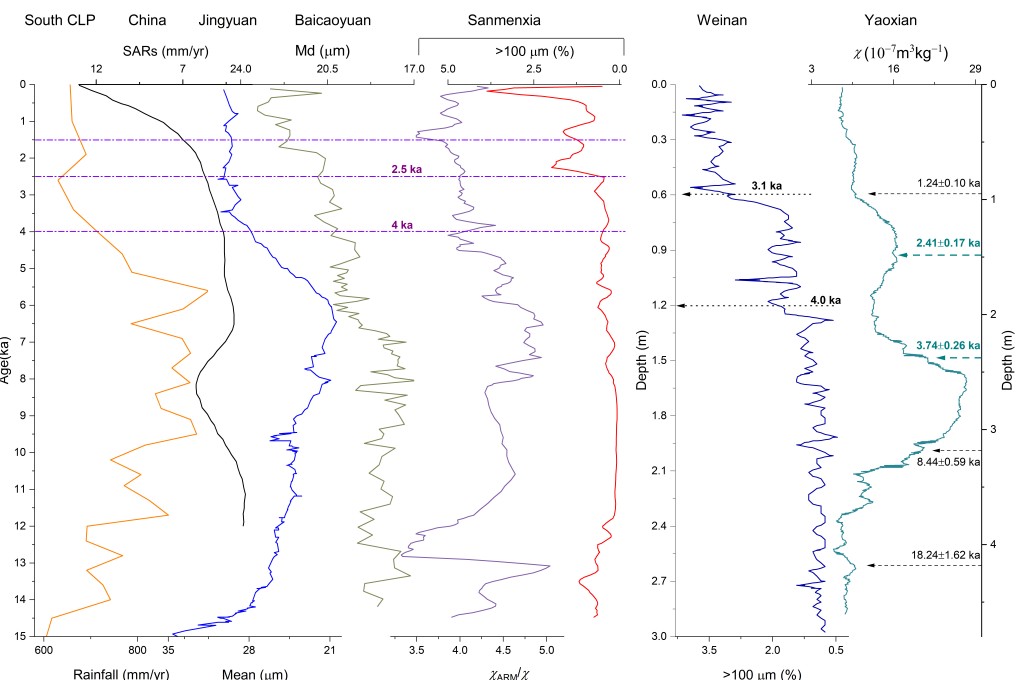

**Figure 11.** Comparisons of variations of multiple proxies from Holocene records. South CLP [10]Be-based rainfall reconstruction (earthy yellow curve) is modified after [53]; synthesized trend of sediment accumulation rates (SARs) from 191 sediment archives in China (black curve) after [5]; mean grain size of Jingyuan loess ( blue curve) after [57]; median grain size (Md) of Baicaoyuan loess (celadon curve) after [26]; $\chi_{ARM}/\chi$ (purple curve) and >100 μm (red curve) of Sanmenxia; >100 μm fraction of Weinan loess (dark blue curve) after [9]; and $\chi$ of Yaoxian loess (cyan curve) after [44].

### 4.2. Human Impacts Overwhelming Climate Control of Soil Erosion on the CLP

The study on the Holocene loess at Guanzhong, south CLP shows that three end members of the loess deposits can be identified [55]. The fine grain end member is interpreted as being associated with weathering and pedogenesis dominated by the East Asia summer monsoons. The coarse grain end member is suggested to originate from near-source sediments from the Yellow River floodplains transported by strong winds. The third end member is thought to be the major component of the loess deposits linked to the northwest monsoon. The study on Mangshan loess suggests that the clayey loess component representing the fine dust supplied over the entire CLP by long-term suspension processes during major dust outbreaks and as part of a background supply system [65]. The Sanmenxia loess in southeast CLP partly sources from the middle reaches of the Yellow River [66]. Therefore, the changes in the magnetic parameters of loess from southwest CLP can be partly attributed to the overall weathering and erosion variation across the CLP [67,68].

The adjacent Wangguan loess section is notable for its high sedimentation rate in the last loess–paleosol couplet [29]. The accelerated loess accumulation is attributed to the more fluvial sediments serving as a proximate dust source produced by the intensified incision of the Yellow River. However, the Holocene loess at Wangguan is very thin. Similarly, the Holocene loess at Mangshan is thin and shows no increase in accumulation rate in the middle-late Holocene, though it is also characterized by the major contribution of the floodplains to loess accumulation [69]. In the Sanmenxia section, the <63 μm fraction show little change at 3.5–1 m (9.2–2.5 ka), suggesting that the changes in proximate source areas is limited. Besides, the persistent changing trend in various proxies at Sanmenxia indicates that the magnetic behavior and grain size characteristics are not likely caused by occasional overland flows. Thus, significant changes that occurred at 4–5 ka can be interpreted as responding to soil erosion in wide areas rather than local source change. In the cool-dry periods, soils erosion was generally very low because there was low precipitation. In warm-

wet periods the land surface might be protected by a well-developed vegetation cover (forest or steppe), which is not favorable to soil erosion. In general, soil erosion during the transition from cool-dry to warm-wet stages was more intensive than that during the warm-wet to cool-dry transition [11]. The synthesized study suggests that 5–6 ka, when the warm-wet Holocene optimal was turning into the cool-dry late Holocene, was a transitional period with anthropogenic impacts intensifying in China at a continent scale [5]. Enhanced human activities through land-use modifications altered vegetated landscapes and resulted in accelerated soil erosion after 5 ka in China [5,6]. Land cover change was pervasive at 5–6 ka across the CLP, as sediment records across the CLP suggest [62]. The 4–6 ka interval shows a noticeable decrease in $\chi_{ARM}/\chi$ and redness and a increase in grain size in the Sanmenxia loess. However, it is hard to partition the natural and anthropogenic contributions. Within chronological uncertainty, 4–6 ka could be regarded as a transitional period before the onset of accelerated soil erosion at Sanmenxia (Figures 10 and 11). The intensified human impact at 4 ka was coeval with the onset of the Bronze Age and the first prehistoric dynasty in China [21,45].

Moreover, the soil erosion at Sanmenxia could be correlated to that in Baicaoyuan, Weinan [9], or Yaoxian [44] (Figure 11). The mean grain size of Jingyuan loess, which is suggested to be sensitive to the Asian summer monsoon intensity, has had low amplitude since 4 ka [57]. Note that Md of Baicaoyuan, which was interpreted to be indicative of monsoon intensity, is more similar to $\chi_{ARM}/\chi$ than to Md of the Sanmenxia loess (Figures 3 and 11).

An increase in coarse grain size fraction in loess is generally interpreted as resulting from the drying climate [41]. However, loess grain size can also be affected by erosion. Previous studies in south CLP suggest that >100 μm fraction indicates human-induced erosion [9]. As revealed by SIRM versus $\chi$ relation, SIRM/$\chi$, and >100 μm fraction (Figures 8d, 10 and 11), soil erosion intensified after 2.5 ka at Sanmenxia. Since 1.5 ka, soil erosion further increased. The rapid changes in $(CaO + Na_2O + MgO)/TiO_2$ at around 1.5 ka are consistent with the intensified soil erosion and weathering intensity at 1.8 ka in southwest China [19]. The low $\chi_{ARM}/\chi$ possibly responds to the weakening hydroclimate. The coarse grain size fraction (>100 μm%) and weak relation between SIRM and $\chi$ indicate aggravated soil erosion, while the geochemical parameters suggest intensified chemical weathering due to soil erosion. A recent study demonstrated that the effect of intensified human activity outpaced natural climatic variability on earth surface processes as the major control of dust storm activity since 2 ka [10], which is consistent with the late Holocene renewed acceleration in erosion on the CLP. Besides, the studies on the Duowa loess section [7,8], Dadiwan marsh section [56], sediment discharge of the Yellow River [11], the sediment yield of the Yellow River (Figure 12) [12–14], and increased fire activity [20] and poor grain production in China [18] also indicate temporal coherence with our results.

Instead of land use cycles or a single significant increase in soil erosion in the middle Holocene (5–6 ka) or in the late Holocene (within the 1.5–3.1 ka span) highlighted in previous studies, two increases in soil erosion were indicated in the Sanmenxia loess, with 4–6 ka being the transitional period before the first increase. Such a temporal pattern is similar to that of the intensification of pastoralism revealed by lake sediments in the northeastern margin of the CLP [15].

### 4.3. Contrasting Sustainability between Zhejiang and CLP

In Zhejiang, the prehistoric culture was suggested to be dominated by climate with the turning point of the response of prehistoric culture to climate change happening at 4 ka [24,25,72,73]. BHQ showed a significant increase in the pollen of cultivated *Oryza* during the environmental evolution from lake to marsh at 5.68–4.17 ka, implying that the cool and dry climate might result in the declining level of groundwater and be conducive to human activity and cultural prosperity in the area [31]. The high content of <4 μm fraction at 0–2.5 m of BHQ2 might result from the weakened hydrodynamics after 4 ka. The study area was less susceptible to human disturbance. There was no acceleration in sediment

accumulation or soil erosion rates in the drying climate since the middle Holocene, as the BHQ cores and other local records indicate [24,25,72].

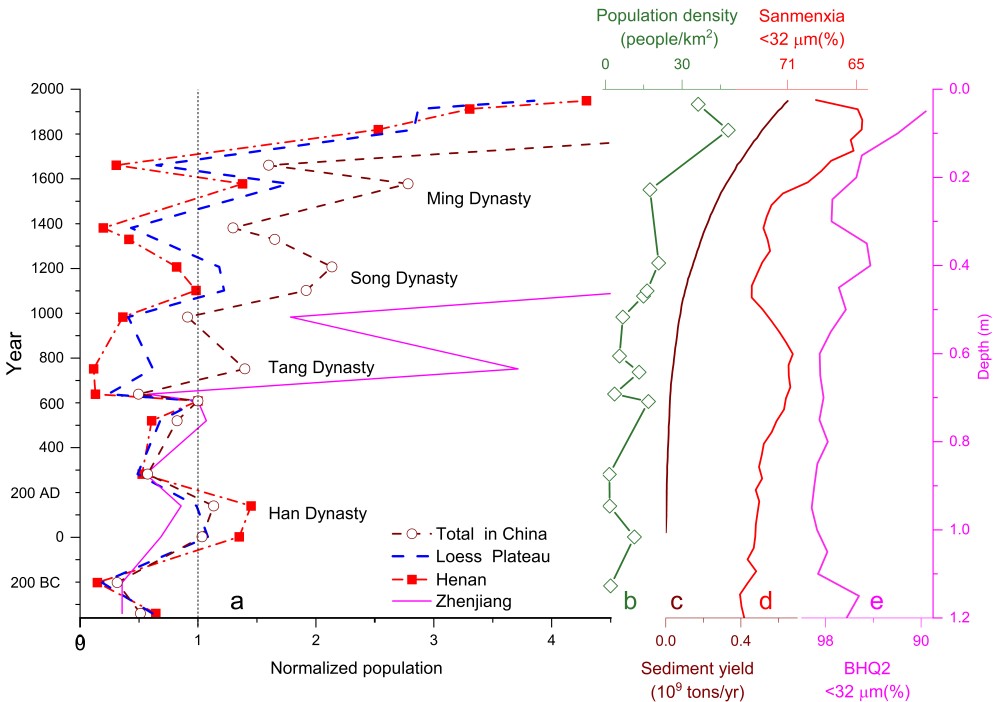

**Figure 12.** a: Population fluctuations of Zhejiang, Henan, Loess Plateau (Henan, Shanxi, Shaanxi, Gansu and Ningxia), and China since 340 BC (normalized by the population in 606 AD) [70]; b: population density in hilly or gully areas on the CLP, modified after [71]; c: anthropogenic increment of sediment yield of the Yellow River since 2 ka, after [12]; d: <32 um fraction of Sanmenxia loess since 2.36 ka; e: <32 um fraction of BHQ2 (0–1.2 m).

With the higher population density than on the CLP since the Early Tang Dynasty (1.4 ka) [70], the present-day Zhejiang Province might see much less human-induced soil erosion. As Figure 12 demonstrates, the population in Henan or on the CLP fluctuated and did not increase significantly until after the Qing Dynasty (1616 AD), while the population in Zhejiang increased dramatically after Tang Dynasty (618 AD). The population growth on the CLP is approximately correspondent with the anthropogenic increment of sediment yield of the Yellow River [12,13,70,71] and the coarsening trend of Sanmenxia loess (Figure 12).

The comparable increasing <32 μm fraction of Sanmenxia and BHQ2 indicate similar drying trends in Zhejiang and the CLP. The contrast in human-induced soil erosion and population growth could be partly attributed to the difference between the loess on the CLP and the zonal soils in Zhejiang. Similar negative feedback between civilization and the environmental impact to those reported elsewhere [4,16] could be traced. The origin of Chinese agriculture is believed to be associated with loess [74]. Natural vegetation helps to enhance the porosity and, thus, the permeability and water-storage capacity of loess. However, human destruction of natural vegetation would result in the initiation of serious soil erosion. On the CLP, grazing was still the major type of land use as late as 2.4 ka [2]. Cultivation and settlement occurred since the Qin dynasty (221 BC). However, much farmland was abandoned from the Western Han Dynasty to the Sui Dynasty (9–618 AD). Farming became pervasive again since the Tang dynasty (618 AD). The resultant accelerated soil erosion, possibly canceling out technology development and affecting land productivity and crop yield.

The CLP was no more the most prosperous area in China, and the population proportion of CLP decreased, while the southeast of China developed rapidly since 1 ka. Given

the enormous disparities across China, understanding the combined effects of geographical setting, soil formation, hydroclimate, and human activity at the regional level is of significance to sustainable development.

## 5. Conclusions

The multidisciplinary study of the Holocene loess on the CLP and the temporal coherence with other records suggest that the effect of intensified human activity outpaced climatic variability on earth surface processes, beginning in the middle Holocene. The first phase of significant human-driven soil erosion probably began at around 4 ka. A renewed acceleration in soil erosion was indicated to occur during the 2.5–1.5 ka span. By contrast, Zhejiang, in the southeast of China, shows much lower susceptibility to soil erosion. The reverse correlation between enhancing soil erosion and relative population growth represents a negative feedback between civilization and the environmental impact on the CLP, where agriculture development resulted in decreased land productivity and undermined regional sustainability.

**Author Contributions:** Conceptualization, Q.C. and G.Z. (Guoyong Zhao); methodology, Q.C. and G.Z. (Guoyong Zhao); software, G.Z. (Guocheng Zhang) and W.W.; validation, Q.C. and G.Z. (Guoyong Zhao); formal analysis, Q.C. and G.Z. (Guoyong Zhao); investigation, Q.C. and G.Z. (Guoyong Zhao); resources, Q.C., J.G. and G.Z. (Guoyong Zhao); data curation, Q.C. and G.Z. (Guoyong Zhao); writing—original draft preparation, Q.C.; writing—review and editing, Q.C. and G.Z. (Guocheng Zhang); supervision, J.G. and Q.C.; project administration, Q.C. and J.G.; funding acquisition, Q.C. All authors have read and agreed to the published version of the manuscript.

**Funding:** This research was funded by the National Natural Science Foundation of China (Grant Nos. 41402155 and 42130507).

**Data Availability Statement:** The data used in this work can be obtained in Figshare (https://doi.org/10.6084/m9.figshare.19520140.v1, accessed on 6 April 2022) or are available from the authors upon request.

**Acknowledgments:** We thank Ye Wei and Zhu Lidong for the fieldwork at Beihuqiao village and the interpretation of the core records and Liu Xiuming and LÜ Bin for their help in magnetic measurements and related resources.

**Conflicts of Interest:** The authors declare no conflict of interest.

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
