# Peer review of "Anthropogenic Influences on Soil Erosion since the Late Holocene and Contrasting Regional Sustainability in China"

_sustainability, doi:10.3390/su14116600_

Round 1
Reviewer 1 Report
Review, Manuscript ID: sustainability-1703115
`Anthropogenic Influences on Soil Erosion since the late Holocene and Contrasting Regional Sustainability in China`
Thank you for the opportunity to review this manuscript.
The technical, structural and formal quality of the manuscript is good but there are some typos which need to be corrected.
Please see the attached pdf file for my suggestions.

Author Response
Dear reviewer:
Thanks a lot for your corrections and suggestion! I have revised the manuscript accordingly. Please see the attached revised manuscipt.
(High- resolution figures were originally provided )
Best regards!
Qu Chen

Reviewer 2 Report
Dear Authors,
The article is very valuable but I do not find a proper connection between the title, the aims and the content.
I mean, you have very detailed data about the upper 7 metres and sometime up to 20 metres and sometimes up to 20 000 years, but, considering the antrophognic effects, I do not really see the reason for this, I do not see the connection and I do not even know the trustworthiness of the data about the size of the population back to 1000 or 2000 years.
You do not even have results related to anthropogenic influences on soil erosion, you have a subchapter in the discussion about it.
I do not see the relevance/meaning of the "Contrasting Regional Sustainability in China" part of the title.
The article is basically a stratigraphic analysis and it would fit much better to a journal about sedimentation, I think.
The manuscript is valuable!
Regards, Reviewer X
Author Response
Dear reviewer:
Many thanks for your valuable comments. I am sorry
that the manuscript was not written well enough. please see the attachment.
I am still think about a new title. I hope that you could see some improvement in the revised manuscript.
Thanks again !
Chen Qu

Reviewer 3 Report
The work deals with very important and interesting subject matter, but the main and most important weakness reducing the value of this work is the lack of clearly formulated objective or research hypothesis.
The introduction should be slightly longer and end with the formulation of the aim or hypothesis.
Please, move the figure from the introduction to materials and methods, as that is where it belongs ( approximately line74).
Line 106 Provide information on what optical parameters of the sample were used when measuring with Mastersizer 2000.
The paper should be closed with a summary or conclusion confirming or refuting the research hypothesis, or referring to the purpose of the paper if no hypothesis has been formulated.
Author Response
Dear reviewer:
Thank you very much for your valuable comments
please see the attachment.
Best regards!
Chen Qu

Round 2
Reviewer 2 Report
Dear Authors,
Thank you for your detailed answers.
I think the manuscript is publishable now.
Regards, Reviewer X
Reviewer 3 Report
-